# Chemo-Enzymatic Synthesis and Biological Assessment of *p*-Coumarate Fatty Esters: New Antifungal Agents for Potential Plant Protection

**DOI:** 10.3390/molecules28155803

**Published:** 2023-08-01

**Authors:** Cyrian Thaeder, Juliette Stanek, Julien Couvreur, Célia Borrego, Fanny Brunissen, Florent Allais, Amandine L. Flourat, Sylvain Cordelier

**Affiliations:** 1URD Agro-Biotechnologies Industrielles (ABI), Centre Européen de Biotechnologies et Bioéconomie (CEBB), AgroParisTech, 3 rue des Rouges Terres, 51110 Pomacle, France; cyrian.thaeder@utc.fr (C.T.); jcouvreur86@gmail.com (J.C.); fanny.brunissen@agroparistech.fr (F.B.); florent.allais@agroparistech.fr (F.A.); 2UFR Sciences Exactes et Naturelles, Université de Reims Champagne Ardenne, RIBP EA 4707, USC INRAE 1488, 51100 Reims, France; juliette.stanek@univ-reims.fr (J.S.); celia.borrego@univ-reims.fr (C.B.)

**Keywords:** *p*-coumaric acid, chemo-enzymatic synthesis, coumarate fatty esters, Sclerotiniaceae, rapeseed, antifungal, plant protection

## Abstract

One trend in agriculture is the replacement of classical pesticides with more ecofriendly solutions, such as elicitation, which is a promising approach consisting of stimulating the natural immune system of a plant to improve its resistance to pathogens. In this fashion, a library of *p*-coumaric-based compounds were synthesized in accordance with as many principles of green chemistry as possible. Then, these molecules were tested for (1) the direct inhibition of mycelium growth of two pathogens, *Botrytis cinerea* and *Sclerotinia sclerotiorum*, and (2) plasma membrane destabilization in Arabidopsis and rapeseed. Finally, the protective effect was evaluated on an Arabidopsis/*B. cinerea* pathosystem. Total inhibition of the growth of both fungi could be achieved, and significant ion leakage was observed using dihydroxylated fatty *p*-coumarate esters. A direct effect on plants was also recorded as a ca. three-fold reduction in the necrosis area.

## 1. Introduction

The switch from a petro- to a bio-economy will increase the demand for biomass. In addition, climate change regularly increases stresses that threaten crops (e.g., drought, pests), leading farmers to adapt their agricultural practices for maintaining productivity. It is, therefore, a necessity to develop sustainable treatments to protect crops against these stresses to maintain their yield. Moreover, *Botrytis cinerea* and *Sclerotinia sclerotiorum*, fungi from the Sclerotiniaceae family, are widespread plant pathogens with a necrotrophic lifestyle that causes diseases in many crops (more than 200 plant species), such as fruit or oil crops (e.g., grapevine or rapeseed) [1,2,3,4]. They cause a lot of damage to crops and can reduce yield drastically, leading to huge economic losses. To prevent this, fungicides are usually applied to crops [4,5]. However, these types of chemical pesticides can be detrimental to human and environmental health; therefore, there is a real need to find some alternative practices to avoid the development of this pathogen without the side-effects of fungicides [6].

Commonly used chemical fungicides generally display direct antifungal activity by inhibiting the germination or the growth of fungi. New bio-based molecules have already shown this direct antifungal activity [7,8]. However, the stimulation of the natural plant immune system seems to be an interesting approach to prevent fungal damage and consequently to reduce important yield loss [9]. The stimulation of plant immunity, generally called elicitation, goes through the perception mechanism that occurs at the plasma membrane level, leading to the induction of early plant defense signaling, such as extracellular ion leakage and reactive oxygen species (ROS) production [10]. This early signaling could then trigger the production of plant defense compounds, potentially leading to plant disease protection [6,11]. Recently, amphiphilic surface-active molecules, such as rhamnolipids and lipopeptides, produced by a variety of microorganisms, have been studied in the context of plant protection through immune system stimulation [6]. These compounds have similar dual effects of protecting plants through antifungal activity and stimulating local and/or systemic plant immunity. Although numerous elicitors are perceived by plasma membrane receptors, recent studies on these amphiphilic compounds suggest that they are sensed in an uncommon way involving lipids in the bilayer of the plant plasma membrane [12,13]. It seems that the amphiphilic properties of these compounds could explain their singular elicitor activity [14,15].

Interestingly, *p*-coumaric acid, a natural cinnamic acid present in numerous plants, and its esters derivatives exhibit antifungal properties [11,16]. While this compound can be extracted from numerous agricultural by-products [17,18], its bioproduction from engineered strains is also demonstrated and could unlock some purification drawbacks [19,20]. Moreover, it has already been established that alkyl chains with at least eight carbons can efficiently destabilize fungal cell membranes, leading to moderate antifungal activity. In addition, this chain length has also been proven to interact with the plant cell’s plasma membrane and could induce immune responses [6,8,21]. These chains can be introduced in the *p*-coumaric acid scaffold using various methodologies. Herein, two green pathways were evaluated based on our previous works dealing with solvent-free enzymatic transesterification [22,23,24] and proline-mediated Knoevenagel-Doebner condensation [25,26]. Further modifications could be performed to increase the amphiphilic properties of *p*-coumarate fatty esters to enhance their biological activities. The aims of this study are to (i) use naturally occurring *p*-coumaric acid as a platform molecule, (ii) functionalize it through chemo-enzymatic pathways with carbon chains of various lengths in order to obtain agro-sourced products, and (iii) evaluate its potential as a plant protection compound against *B. cinerea* and *S. sclerotiorum*. Herein, a library of alkyl *p*-coumarates were synthesized and further functionalized prior to the investigation of the synergy between coumarate nucleus and fatty alcohol chains against the pathogenic fungi *B. cinerea* and *S. sclerotiorum* by measuring direct growth inhibition. The elicitation effect of the synthesized compounds was also evaluated on Arabidopsis and rapeseed plants through plasma membrane destabilization. Finally, the most promising candidates were tested on an Arabidopsis/*B. cinerea* pathosystem to demonstrate their putative interest in plant protection.

## 2. Results and Discussion

### 2.1. Synthesis

#### 2.1.1. Linear Alkyl Coumarate Esters

This study started with the synthesis of a library of alkyl *p*-coumarates with fatty alkyl chains (octyl to tetradecyl (C8–C14)) using two methodologies. The first one relied on the lipase-mediated transesterification of ethyl *p*-coumarate **1** with fatty alcohols [27,28], whereas the second one considered the proline-mediated Knoevenagel-Doebner condensation of *p*-hydroxybenzaldehyde with mono-malonate esters leading to molecules **2** to **8** (Figure 1) [25,29]. The global yield achieved with the first method ranged between 53% and 59%, whereas the second method provided the desired compound in 40–53% yield by applying the conditions described by Peyrot et al. [25]. In this previous publication, the moderate yield for the Knoevenagel-Doebner condensation of the *p*-hydroxybenzaldehyde had already been reported because of the low activation of the aldehyde moiety. In comparison, the enzymatic pathway led to slightly higher yield even if the presence of the double bond decreasing the reactivity [27].

To achieve a higher yield, the Knoevenagel-Doebner condensation conditions were slightly adapted. The L-proline and 4-hydroxybenzaldehyde amounts were increased to 1 and 1.5 equivalent, respectively, compared to the mono-alkyl malonate, allowing the recovery of dodecyl *p*-coumarate **6** in 72% yield. For the enzymatic pathway, the recovered yield of **6** could be improved from 58% to 77% by switching from normal to reverse phase flash chromatography. Indeed, in normal phase, a portion of the desired product was eluted with the remaining fatty alcohol, thus lowering the recovered yield.

#### 2.1.2. Ramified Coumarate Esters

A second series of coumarate esters was synthesized by using a fatty alcohol ramified in C2 to improve the interaction of the molecules with the lipidic membrane of the fungi or that of the plant cell [14]. The 2-Butyl octanol, 2-hexyl decanol, and 2-decyl dodecanol were grafted onto the *p*-coumarate synthon using the Knoevenagel-Doebner condensation-affording compounds **9** to **11**, respectively (Figure 1). Indeed, our attempts of using the enzymatic pathways proved unsuccessful despite the presence of a primary alcohol. The steric hindrance was hypothesized to be responsible for the lack of reactivity with the enzyme.

To achieve an ester linkage between the two fatty chains, we first thought about a regioselective enzymatic transesterification of ethyl *p*-coumarate with the primary alcohol of 1,2-decanediol, followed by the introduction of a fatty acid to esterify the secondary alcohol. Our group already reported such a selectivity using glycerol and ethyl ferulate [23]. Unfortunately, no reaction was observed between ethyl *p*-coumarate and 1,2-dec-nediol. However, we have already observed that 1,2-decanediol can react with ethyl dehydroferulate using CAL-B. A previous study has evidenced that ethyl dehydro-*p*-coumarate was less reactive toward the enzymatic transesterification than the ferulate or sinapate moieties [27]. In addition, the presence of a double bond also reduces its reactivity. For the transesterification using ethyl dihydroxy-*p*-coumarate (*vide infra*) with 2-butyl octanol, 40% yield of the desired product was obtained. The aforementioned results demonstrate that the double bond of ethyl coumarate prevents the Novozym 435-catalyzed transesterification with ramified fatty alcohols.

Another strategy was attempted to produce the *p*-cinnamate esters with branched fatty esters possessing an internal ester moiety. Decanoic acid and decane-1,2-diol were thus esterified in presence of Novozym 435, then the resulting intermediate 2-hydroxydecyl decanoate **12** was reacted with Meldrum’s acid to give a mono-malonate ester. The latter finally underwent a Knoevenagel-Doebner condensation with 4-hydroxybenzaldehyde to provide the desired ramified diester **13** in 17% global yield (Figure 2).

#### 2.1.3. Improvement of the Hydrophilicity

Even if the ability to interact with the lipidic membrane is required to achieve elicitation, the molecules that exhibit such ability already described in the literature are mainly amphiphilic, such as rhamnolipids [12,13].

Two strategies were investigated to increase the hydrophilicity of the novel aforementioned molecules: (i) addition of a sugar moiety, and (ii) dihydroxylation of the double bond of the coumarate moiety.

##### Glucosylation

The glucosylated analogs, our first targets, can be obtained through various pathways by combining glucosylation and Knoevenagel-Doebner condensation or transesterification (Figure 3). Glucosylated *p*-hydroxybenzaldehyde **14**, ethyl *p*-coumarate **15**, and dodecyl *p*-coumarate **16** were obtained in 23%, 40%, and 30% yield, respectively, by reacting the corresponding phenolate with bromo-2,3,4,6-acetoxy glucose using the procedure described by Ferrari et al. [30]. From **14**, Knoevenagel-Doebner condensation led to a low yield (21%). Under our conditions, the elongation of the alkyl chain through enzymatic transesterification was also unsuccessful due to the immiscibility between **15** (or **17**) and 1-dodecanol. From all possible pathways, the desired product **18** was obtained in acceptable yield (22% global yield) only through the direct glucosylation of dodecyl *p*-coumarate followed by the deacetylation.

##### Dihydroxylation

Inspired by the structure of the 3-hydroxy-fatty acid chain of the rhamnolipids, we decided to design *p*-coumarate esters possessing hydroxy groups on the α and β positions of the carbonyl moiety through the dihydroxylation of the double bound of the corresponding *p*-coumarate derivatives (Figure 4). From the preliminary attempts, it was found that an acetylation of the free phenol of the coumarate esters had to be performed before conducting the dihydroxylation. This was successfully achieved to afford compounds **19** to **22** in excellent yield (>95%). The modified Upjohn dihydroxylation procedure was performed to efficiently dihydroxylate the acetylated coumarate esters [31,32]. The data proved that the efficiency of the dihydroxylation seems correlated to the chain length of the alkyl *p*-coumarate esters. Indeed, 75% yield was achieved for compound **23** from ethyl *p*-coumarate whereas 69%, 55%, and 50% yields were recorded for the octyl, decyl, and dodecyl *p*-coumarates, respectively, after the deprotection of the acetate group (compounds **24** to **26**). It is noteworthy to mention that the longer alkyl chain can be obtained for the dihydroxylated series by the enzymatic transesterification of **23** in satisfying yield (65%). Thus, this divergent pathway appears more efficient as it allowed for the synthesis of esters with all chain lengths in good yield from a unique precursor. In addition, the transesterification of **23** into **28** with 2-butyl octanol was achieved in 40% yield.

### 2.2. Biological Activities of p-Coumaric Acid-Based Molecules

#### 2.2.1. Direct In Vitro Antifungal Activity against *B. cinerea* and *S. sclerotiorum*

All newly synthesized molecules were tested against *B. cinerea* and *S. sclerotiorum* to evaluate their direct antifungal effect at 100 µM, including *p*-coumaric acid as a control. The results are expressed as the relative growth inhibition of the fungal mycelium on PDA medium (Figure 2). The *p*-coumaric acid (**0**) used as the platform molecule showed a relative growth inhibition of ca. 25–30% for both fungi. When it was functionalized with a single carbon chain of variable length, the relative growth inhibition against *B. cinerea* (Figure 2A and Figure A1) increased from 35% to 80% for the 10-, 9-, and 8-carbon chain length, respectively (**4**, **3**, and **2**). The relative growth inhibition is then lower than that of the *p*-coumaric acid alone for longer chain length (11- to 14-carbon length, compounds **5**–**8**). The addition of a second carbon chain to the platform molecule also reduced the relative growth inhibition (**9**–**11**) except for the 10-10-double carbon chain (**13**) that increased the relative growth inhibition by up to 100% against *B. cinerea* and 50% against *S. sclerotiorum* (Figure 2A,B). Interestingly, when the *p*-coumaric acid was functionalized with a single carbon chain and dihydroxylated, the relative growth inhibition increases by up to 100% (**24**–**28**). The molecule with a 12-carbon chain (**26**) was less effective than the other dihydroxylated coumarates against *B. cinerea* (82% of relative inhibition against 100% for the other molecules).

These results show that *p*-coumaric acid functionalized with a short single carbon chains (8 to 10 carbons) or with a double carbon chain of 10-10 carbon displays a higher antifungal activity against *B. cinerea* and *S. sclerotiorum* mycelium. The antifungal effect against both fungi was even higher with the dihydroxylated compounds (100% relative inhibition at 100 µM) compared to *p*-coumaric acid alone (30% relative inhibition at 100 µM). These results suggest that adding a fatty ester to the dihydroxylated *p*-coumaric acid enhances its insertion into the fungal membrane. They also highlight the importance of the carbon chain length, the optimal length being between 8 and 10 carbons. These results are in accordance with previous studies that used rhamnose as a platform molecule [8,21].

#### 2.2.2. Destabilization of the Plant Plasma Membrane

The electrolyte leakage detection is often used as a basic tool to determine a stress response in plant cells [33]. Indeed, a physical or chemical stress of a plant cell, such as the insertion of a molecule in the plasma membrane, can lead to a destabilization of the lipidic structure of the plasma membrane, triggering extracellular ion leakage. The ability of the molecules to destabilize the plasma membrane of plant cells was determined by ion leakage assay in Arabidopsis leaves and rapeseed cotyledons. The ion leakage quantification was performed directly after the treatment (0 hpt) and 48 h after (48 hpt). The results expressed as the electrolyte leakage (in µS/cm) are displayed in Figure 3.

The *p*-coumaric acid (**0**) induced an electrolyte leakage of around 100 µS/cm at 48 hpt compared to 15 µS/cm for the non-treated sample (NT) in both the Arabidopsis and rapeseed samples. When it was functionalized with a single carbon chain of variable length, the electrolyte leakage in Arabidospsis leaves was at a similar level, respectively for the 10-, 9-, and 8-carbon chain length (**4**, **3**, and **2**, respectively) (Figure 3A). The electrolyte leakage was then lower than that of *p*-coumaric acid alone for longer chain length (11- to 14-carbon length, **5** to **8**). The addition of a second carbon chain to the platform molecule also reduced the electrolyte leakage (**9** to **13**). Interestingly, when the single chain *p*-coumaric ester was dihydroxylated, the electrolyte leakage increased up to 300 µS/cm (**24** to **28**). The molecule with a 12-carbon chain (**26**) is still less effective than the other dihydroxylated compounds (200 µS/cm). On rapeseed cotyledons, the results are very similar to that of Arabidopsis leaves (Figure 3B) except for the molecules with double carbon chains, **9** and **10,** for which the electrolyte leakage increased up to 300 µS/cm. As seen in the previous Figure 2, dihydroxylated molecules seem to have the most destabilizing effect on the plant cell’s plasma membrane (approximatively 300 µS/cm compared to 100 µS/cm for *p*-coumaric acid). 

These results confirm those observed previously on the direct antifungal activity. They suggest that the dihydroxylation of the single carbon chain *p*-coumarate ester improves the molecule insertion into the plant plasma membrane, leading to extracellular ion leakage in the plant cell or growth inhibition of fungal mycelium.

#### 2.2.3. Arabidopsis Protection against *B. cinerea*

The ability of the dihydroxylated molecules to protect the Arabidopsis plant against *B. cinerea* was studied. The Arabidopsis plants were treated with molecules at 100 µM and were placed in growth chambers. Three days later, three to four leaves of these Arabidopsis plants were inoculated with 5 µL of a *B. cinerea* conidia suspension (10^5^/mL) on the center of each leaf. The necrosis areas were measured 3 days post-inoculation to calculate the relative inhibition of *B. cinerea* growth (Figure 4). 

All the dihydroxylated molecules did not induce the same protection against the pathogen in the plant. Actually, the necrosis on plants treated with compound **25** is three times smaller than that on the negative control, which makes it the most effective in this case (57% of necrosed surface compared to 20%, respectively). This result can be explained by the significant direct antifungal effect of this molecule against *B. cinerea* (Figure 2A). Moreover, it is also the most effective in electrolyte leakage (Figure 3A). In this case, the 10-carbon chain length was the most effective against this pathogen. However, all the dihydroxylated molecules were more effective against the pathogen than the negative control. The least effective was the one with an eight-carbon chain length (**24**). The others with a 12-carbon chain length (compound **26** and **28**) were equally efficient (31% and 34% of necrosed area, respectively). Similarly to previous studies [8,21], our results demonstrate that the synthetic molecules displaying amphiphilic properties can induce plant protection. Robineau et al. found that synthetic smRLs with an acyl chain of 10 carbons at 300 µM were able to increase the resistance of tomato plants against *B. cinerea* in controlled conditions [8]. Interestingly, Platel et al. shown that synthetic RLs with a 12-carbon fatty acid tail were the most effective at 3200 µM for protection efficacy on the wheat-*Zymoseptoria tritici*—pathosystem [21]. Our results also demonstrate the importance of the length of the carbon chain. In our case, it seems that the dihydroxylated 10-carbon chain triggers higher Arabidopsis protection against *B. cinerea* at a lower concentration of 100 µM. The dihydroxylated *p*-coumaric compounds with single carbon chains of 10 or 12 carbons seem to be good candidates to be used in biocontrol strategies to protect plant against fungal diseases. 

### 2.3. Discussion and Perspectives

Herein, various strategies have been explored to synthesize target compounds. The data on biological activities allowed for the identification of promising candidate molecules for crops protection. However, the optimization of the synthetic pathway is required to make it greener and more efficient. The optimization of the transesterification procedure can be conducted to avoid the last purification. Nevertheless, the main drawback of the current pathway is the dihydroxylation step, and further investigations are needed to find a less toxic catalyst. It was noticed that during our investigations, an enzymatic epoxidation [34,35], followed by a subsequent hydrolysis, was attempted unsuccessfully. The chemical alternatives to osmium catalysts have been reviewed by Bataille et al. [36]. More recently, Wei and co-workers have developed an iron-mediated dihydroxylation with aqueous H_2_O_2_ [37]. In addition, the enzymatic solutions start to emerge [38,39]. The latter will require more time to be implemented but could avoid the protection/deprotection of the phenolic moiety. A step further could also be the insertion, in a single engineered strain, of the *p*-coumaric acid pathway and the dihydroxylation. In summary, a screening of existing chemical alternatives could be performed to achieve rapidly greener conditions, while a deep investigation should be conducted to allow for enzymatic dihydroxylation.

These new dihydroxylated *p*-coumaric esters with a 10- to 12-carbon chains appear to be potentially good candidates for plant protection against fungal diseases. These compounds seem to share similar properties with amphiphilic biosurfactants, such as glycolipids or lipopeptides produced by microorganisms. Indeed, similarly to dihydroxylated *p*-coumaric esters, they display a direct antifungal activity and are able to perturbate the plant plasma membrane, triggering a plant protection response against fungal phytopathogens. Since these biosurfactants are purified from microorganism growth media, their costs, their efficacies in the field, and their purity have to be improved to allow them to be used at a higher degree in crop protection [6]. With the improvement and the optimization of the synthetic pathway of the dihydroxylated *p*-coumaric esters, these compounds could have a lower cost production with a higher purity. Moreover, since these dihydroxylated *p*-coumaric esters show biological activities at low concentrations (e.g., EC_50_ (decyl dihydroxycoumarate (n°25)) = 25 µM against fungal phytopathogen *B. cinerea* (Figure A2)), they could be very good candidates for in-depth complementary studies to decipher the perception and the mechanisms of action of amphiphilic compounds in the induction of plant immunity. For example, further investigations could analyze their ability to alter the physicochemical properties of the bilayer and to permeabilize membranes that could result in their lysis [40]. More detailed studies would also be required to characterize their capacity to induce hallmarks of elicitor-triggered plant immunity, such as the accumulation of ROS, calcium influx, phosphorylation cascade, callose deposition, or defense gene activation [41,42,43]. Finally, given their valuable properties, it would be interesting to evaluate their ability to induce disease protection on various crops to really consider their use as biocontrol solutions in integrated pest management.

## 3. Materials and Methods

### 3.1. Material

The reagents and solvents are purchased from various suppliers as indicated in Table 1.

All reactants and solvents were used as received. The purifications using flash chromatography were performed on a PuriFlash 420XS, Interchim (Montluçon, France), equipped with a prepacked column (PF-30SI-HP (silica) or PF-30C18HP (C-18 grafted silica)) and monitored at λ = 254 and 280 nm. ^1^H-NMR analyses were recorded at 300 MHz on a Fourier 300, Bruker (Wissenbourg, France). The spectra were calibrated on the solvent residual peak (CDCl_3_, *δ* = 7.26 ppm; acetone-*d*_6_, *δ* = 2.05 ppm; CD_3_OD, *δ* = 3.31 ppm) and described as follows: chemical shift in part per million (ppm), multiplicity, coupling constant, integration, and attribution. ^13^C-NMR analyses were recorded at 75 MHz on a Fourier 300, Bruker (Wissenbourg, France). The spectra were calibrated on the solvent residual peak (CDCl_3_, *δ* = 77.16 ppm; acetone-*d*_6_, *δ* = 29.84 ppm; CD_3_OD, *δ* = 49.00 ppm) and described as follows: chemical shift in part per million (ppm) and attribution. 2D spectra, ^1^H-^1^H-Cosy, ^1^H-^13^C-HSQC, and ^1^H-^13^C HMBC were used for attributing peaks. HRMS were recorded on a HPLC system Agilent 1290 coupled with PDA UV and 6545 Q-Tof. The melting points were recorded on a MP50, Mettler Toledo (Viroflay, France) using capillary tubes (ME-18552) with the following method: initial temperature 40 °C, then heating at 5 °C/min until 200 °C.

### 3.2. Methods

#### 3.2.1. Transesterification Pathway (A)

##### Synthesis of Ethyl Coumarate

*p*-Coumaric acid (7.0 g, 42.6 mmol) was dissolved in ethanol (106 mL, C = 0.4 M) and 8 drops of concentrated HCl were added. The reaction was refluxed overnight, cooled to room temperature, and concentrated. The crude oil was diluted into ethyl acetate, washed thrice with saturated aq. NaHCO_3_ solution, then with water and brine, dried over anhydrous MgSO_4_, filtered, and concentrated. 7.6 g (92%) of **1** was recovered.

**m.p.** (°C): 79.6. **^1^H NMR** (300 MHz, Acetone-*d*_6_) δ (ppm) = 8.89 (s, 1H, OH), 7.60 (d, *J* = 16.0 Hz, 1H, H5), 7.55 (d, *J* = 8.6 Hz, 2H, H2), 6.90 (d, *J* = 8.4 Hz, 2H, H3), 6.34 (d, *J* = 15.9 Hz, 1H, H6), 4.18 (q, *J* = 7.2 Hz, 2H, H8), 1.27 (t, *J* = 7.1 Hz, 3H, H9).^**13**^**C NMR** (75 MHz, Acetone-*d*_6_) δ (ppm) = 167.4 (C7), 160.5 (C4), 145.1 (C5), 130.9 (C2), 127.0 (C1), 116.7 (C3), 115.7 (C6), 60.5 (C8), 14.6 (C9). **HRMS** (*m*/*z*) [M + H]^+^ 193.0859, found 193.0862.

##### General Procedure for Enzymatic Transesterification

Ethyl *p*-coumarate (1 equiv) and desired fatty alcohol (1.5 equiv) were heated at 75 °C under reduced pressure (50 or 150 mbar depending of fatty alcohol boiling point). When the ethyl p-coumarate was melted, Novozym 435 (10% *w*/*w*) was added and the reaction was pursued under gentle stirring for 24 h. Then, reaction was cooled down to room temperature, diluted in acetone, and filtered to remove Novozym 435 that was rinsed with acetone. Filtrate was concentrated. Crude mixture was purified by reverse phase flash chromatography 80:20 MeOH/water, then 100% MeOH.

#### 3.2.2. Knoevenagel-Doebner Pathway (B)

Meldrum’s acid (1.2 equiv) and fatty alcohol were heated at 85 °C or 95 °C without solvent for 3 h. After cooling down to room temperature, the crude mixture was directly engaged into the next step.

4-Hydroxybenzaldehyde (1.5 equiv) and L-proline (1 equiv) were added to the mono-alkyl malonate dissolved in ethanol (C = 0.5 M in aldehyde). Reaction medium was stirred at 80 °C overnight, then concentrated. The crude was purified by reverse phase flash chromatography 80:20 MeOH/water, then 100% MeOH.

Yields for each compound depending of the used pathway were reported in Table 2.

Octyl coumarate (**2**), white powder, **m.p.** (°C): 72.2. **^1^H NMR** (300 MHz, CDCl_3_) δ (ppm) = 7.63 (d, *J* = 16.1 Hz, 1H, H5), 7.41 (d, *J* = 8.5 Hz, 2H, H2), 6.86 (d, *J* = 8.2 Hz, 2H, H3), 6.30 (d, *J* = 15.9 Hz, 1H, H6), 4.20 (t, 2H, H8), 1.70 (p, 2H, H9), 1.37–1.23 (m, 10H, H1014), 0.88 (t, 3H, H15). **^13^C NMR** (75 MHz, CDCl_3_) δ (ppm) = 168.2 (C7), 158.1 (C4), 144.8 (C5), 130.0 (C2), 127.0 (C1), 115.9 (C3), 115.3 (C6), 64.9 (C8), 31.8, 29.3, 29.2, 28.7, 26.0, 22.7 (C9–14), 14.11 (C15). **HRMS** (*m*/*z*) [M + H]^+^ 277.1798, found 277.1796

Nonyl coumarate (**3**), white powder, **m.p.** (°C): 53.5. **^1^H NMR** (300 MHz, CDCl_3_) δ (ppm) = 7.63 (d, *J* = 16.0 Hz, 1H, H5), 7.43 (d, *J* = 8.7 Hz, 2H, H2), 6.85 (d, *J* = 8.7 Hz, 2H, H3), 6.30 (d, *J* = 16.0 Hz, 1H, H6), 5.60 (s, 1H, OH), 4.19 (t, *J* = 6.7 Hz, 2H, H8), 1.69 (p, 2H, H9), 1.45–1.16 (m, 13H, H10–15), 0.88 (t, 3H, H16). **^13^C NMR** (CDCl_3_, 75 MHz) δ (ppm) = 167.9 (C7), 157.8 (C4), 144.5 (C5), 130.1 (C2), 127.4 (C1), 116.0 (C3), 115.8 (C6), 64.9 (C8), 32.0, 29.6, 29.4, 29.4, 28.9, 26.1, 22.8 (C9–15), 14.3 (C16). **HRMS** (*m*/*z*) [M + H]^+^ 291.1955, found 291.1957.

Decyl coumarate (**4**), **m.p.** (°C): 70.2. **^1^H NMR** (300 MHz, CDCl_3_) δ (ppm) = 7.64 (d, *J* = 16.0 Hz, 1H, H5), 7.41 (d, *J* = 8.3 Hz, 2H, H2), 6.88 (d, *J* = 8.3 Hz, 2H, H3), 6.30 (d, *J* = 15.9 Hz, 1H, H6), 4.21 (t, *J* = 6.7 Hz, 2H, H8), 1.70 (p, *J* = 6.7 Hz, 2H, H9), 1.43–1.17 (m, 14H, H10–16), 0.89 (t, 3H, H17). **^13^C NMR** (75 MHz, CDCl_3_) δ (ppm) = 168.6 (C7), 158.5 (C4), 145.2 (C5), 130.2 (C2), 126.9 (C1), 116.1 (C3), 115.2 (C6, 65.2 (C8), 32.0, 29.7, 29.4, 29.4, 28.8, 26.1, 22.8 (C9–16), 14.2 (C17). **HRMS** (*m*/*z*) [M + H]^+^ 277.1798, found 277.1796.

Undecyl coumarate (**5**), **m.p.** (°C): 62.0.^**1**^**H NMR** (300 MHz, CDCl_3_) δ (ppm) = 7.62 (d, *J* = 15.9 Hz, 1H, H5), 7.43 (d, *J* = 8.7 Hz, 2H, H2), 6.85 (d, *J* = 8.6 Hz, 2H, H3), 6.30 (d, *J* = 16.0 Hz, 1H, H6), 5.59 (s, 1H, OH), 4.19 (t, *J* = 6.7 Hz, 2H, H8), 1.68 (d, *J* = 6.7 Hz, 2H, H9), 1.33–1.23 (m, 16H, H10–17), 0.87 (t, *J* = 7.0 Hz, 3H, H18).^**13**^**C NMR** (CDCl3, 75 MHz) δ (ppm) = 167.8 (C7), 157.7 (C4), 144.4 (C5), 130.0 (C2), 127.3 (C1), 115.9 (C3), 115.7 (C6), 64.8 (C8), 31.9, 29.6, 29.6, 29.6, 29.4, 29.3, 28.7, 26.0, 22.7 (C9–17), 14.1 (C18). **HRMS** (*m*/*z*) [M − H]^−^ 317.2122, found 317.2121.

Dodecyl coumarate (**6**), white powder, **m.p.** (°C): 78.1. **^1^H NMR** (300 MHz, CDCl_3_) δ (ppm) = 7.64 (d, *J* = 16.0 Hz, 1H, H5), 7.41 (d, *J* = 8.4 Hz, 2H, H2), 6.88 (d, *J* = 8.3 Hz, 2H, H3), 6.30 (d, *J* = 15.9 Hz, 1H, H6), 4.20 (t, *J* = 6.7 Hz, 2H, H8), 1.70 (p, *J* = 6.5 Hz, 2H, H9), 1.39–1.20 (m, 18H, H10–17, 18), 0.86 (t, 3H, H19). **^13^C NMR** (CDCl_3_, 75 MHz) δ (ppm) = 168.5 (C7), 158.5 (C4), 145.1 (C5), 130.2 (C2), 126.9 (C1), 116.1 (C3), 115.2 (C6), 65.1 (C8), 32.0, 29.8, 29.8, 29.7, 29.7, 29.5, 29.4, 28.8, 26.1, 22.8 (C9–18), 14.25 (C19). **HRMS** (*m*/*z*) [M − H]^−^ 331.2278, found 331.2278.

Tridecyl coumarate (**7**), white powder, **m.p.** (°C): 70.6. **^1^H NMR** (300 MHz, CDCl_3_) δ (ppm) = 7.63 (d, *J* = 16.0 Hz, 1H, H5), 7.42 (d, *J* = 8.6 Hz, 2H, H2), 6.85 (d, *J* = 8.6 Hz, 2H, H3), 6.30 (d, *J* = 16.0 Hz, 1H, H6), 4.19 (t, *J* = 6.7 Hz, 2H, H8), 1.68 (p, 2H, H9), 1.38–1.18 (m, 22H, H10–19), 0.87 (t, 3H, H20). **^13^C NMR** (CDCl_3_, 75 MHz) δ (ppm) = 168.0 (C7), 158.0 (C4), 144.6 (C5), 130.1 (C2), 127.3 (C1), 116.0 (C3), 115.7 (C6), 64.9 (C8), 32.1, 29.8, 29.8, 29.7, 29.7, 29.5, 29.4, 28.9, 26.1, 22.8 (C9–19), 14.3 (C20). **HRMS** (*m*/*z*) [M − H]^−^ 345.2435, found 345.2435.

Tetradecyl coumarate (**8**), white powder, **m.p.** (°C): 85.1.^**1**^**H NMR** (300 MHz, CDCl_3_) δ (ppm) = 7.63 (d, *J* = 15.9 Hz, 1H, H5), 7.43 (d, *J* = 8.7 Hz, 2H, H2), 6.86 (d, *J* = 8.6 Hz, 2H, H3), 6.30 (d, *J* = 15.9 Hz, 1H, H6), 5.88 (s, 1H, OH), 4.20 (t, *J* = 6.7 Hz, 2H, H8), 1.70 (p, *J* = 6.7 Hz, 2H, H9), 1.26 (s, 22H, H10–20), 0.88 (d, *J* = 6.9 Hz, 3H, H21).^**13**^**C NMR** (75 MHz, CDCl_3_) δ (ppm) = 168.0 (C7), 158.0 (C4), 144.7 (C5), 130.1 (C2), 127.3 (C1), 116.0 (C3), 115.7 (C6), 65.0 (C8), 32.1, 29.8, 29.8, 29.7, 29.7, 29.5, 29.4, 28.9, 26.1, 22.8 (C9–20), 14.3 (C21). **HRMS** (*m*/*z*) [M − H]^−^ 359.2591, found 359.2589.

2-butyl octyl coumarate (**9**), colorless oil, **^1^H NMR** (300 MHz, CDCl_3_) δ (ppm) = 7.62 (d, *J* = 16.0 Hz, 1H, H5), 7.43 (d, *J* = 8.7 Hz, 2H, H2), 6.85 (d, *J* = 8.7 Hz, 2H, H3), 6.31 (d, *J* = 16.0 Hz, 1H, H6), 5.77 (s, 1H, OH), 4.10 (d, *J* = 5.7 Hz, 2H, H8), 1.70 (s, 2H, H9), 1.34–1.24 (m, 35H), 0.93–0.82 (m, 9H, H15, H19). **^13^C NMR** (CDCl_3_, 75 MHz) δ (ppm) = 168.0 (C7), 157.8 (C4), 144.4 (C5), 130.0 (C2), 127.2 (C1), 115.9 (C3), 115.7 (C6), 67.4 (C8), 37.4 (C9), 31.9, 31.3, 31.0, 29.7, 29.0, 26.7, 23.0, 22.7(C10–14, C16–18), 14.1, 14.1 (C15, C19). **HRMS** (*m*/*z*) [M − H]^−^ 331.2278, found 331.2278.

2-hexyl decyl coumarate (**10**), colorless oil, **^1^H NMR** (300 MHz, CDCl_3_) δ (ppm) = 7.62 (d, *J* = 16.0 Hz, 1H, H5), 7.44 (d, *J* = 8.4 Hz, 2H, H2), 6.85 (d, *J* = 8.5 Hz, 2H, H3), 6.31 (d, *J* = 16.0 Hz, 1H, H6), 5.65 (s, 1H, OH), 4.10 (d, *J* = 5.7 Hz, 2H, H8), 1.70 (s, 1H, H9), 1.44–1.06 (m, 24H), 0.92–0.83 (m, 6H, H17, H23). **^13^C NMR** (75 MHz, CDCl_3_) δ (ppm) = 168.0 (C7), 157.9 (C4), 144.5 (C5), 130.1 (C2), 127.4 (C1), 116.0 (C3), 115.8 (C6), 67.6 (C8), 37.5 (C9), 32.1, 32.0, 31.5, 30.1, 29.8, 29.7, 29.5, 26.9, 22.8 (C10–16, C18–22), 14.3 (C17, C23). **HRMS** (*m*/*z*) [M − H]^−^ 387.2904, found 387.2902.

2-octyl dodecyl coumarate (**11**), colorless oil, **^1^H NMR** (300 MHz, CDCl_3_) δ (ppm) = 7.61 (d, *J* = 16.0 Hz, 1H, H5), 7.43 (d, *J* = 8.8 Hz, 2H, H2), 6.84 (d, *J* = 8.7 Hz, 2H, H3), 6.30 (d, *J* = 15.9 Hz, 1H, H6), 5.39 (s, 1H, OH), 4.09 (d, *J* = 5.8 Hz, 2H, H8), 1.75–1.56 (m, 2H, 9), 1.41–1.15 (m, 36H), 0.94–0.78 (m, 6H, H19, H27). **^13^C NMR** (75 MHz, CDCl_3_) δ (ppm) = 167.9 (C7), 157.7 (C4), 144.3 (C5), 130.1 (C2), 127.5 (C1), 116.0 (C3, C6), 67.5 (C8), 37.5 (C9), 32.1, 31.5, 30.1, 29.8, 29.5, 26.9, 22.8 (C10–18, C20–26), 14.27 (C19, C27). **HRMS** (*m*/*z*) [M − H]^−^ 443.3530, found 443.3531.

#### 3.2.3. Synthesis of **13**

Decanoic acid (861 mg, 5 mmol); 1,2-docanediol (871 mg, 5 mmol); and Novozym 435 (173 mg, 10% *w*/*w*) were heated at 60 °C under reduced pressure (50 mbar) for 72 h. The reaction was cooled down to room temperature, diluted in acetone, and filtered to remove Novozym 435 that was rinsed with acetone. The filtrate was concentrated. The crude mixture was purified by flash purification over silica gel using 100% cyclohexane, then 95:5 cyclohexane/EtOAc as eluent. 2-hydroxydecyl decanoate **12** was recovered as a white powder (449 mg, 27% yield).

**m.p.** (°C): 55.0. **^1^H NMR** (CDCl_3_, 300 MHz) δ (ppm) = 4.15 (1H, dd, *J* = 11.3, 3.0 Hz, H11), 3.95 (1H, dd, *J* = 11.3, 7.3 Hz, H11’), 3.83 (1H, tq, *J* = 6.9, 3.0 Hz, H12), 2.34 (2H, t, *J* = 7.5 Hz, H2), 1.63 (2H, p, *J* = 7.3 Hz, H3), 1.51–1.40 (2H, m, H13), 1.36–1.15 (24H, m, H4–9, H14–19), 0.87 (6H, t, *J* = 6.3 Hz, H10, H20). **^13^C NMR** (CDCl_3_, 75 MHz) δ (ppm) = 174.1 (C1), 70.1 (C12), 68.6 (C11), 34.3 (C2), 33.4 (C13), 31.9, 29.6, 29.5, 29.4, 29.4, 29.3, 29.3, 29.2, 25.4, 25.0 (C3), 22.7, 14.1 (C10, C20). **HRMS** (*m*/*z*) [M + H]^+^ calculated 329.3050, found 329.3046, [M + Na]^+^ calculated 351.2869, found 351.2868.

Pathway B was used to synthesize **13**. The crude mixture was purified by flash purification over silica gel using 95:5 to 60:40 cyclohexane/EtOAc gradient as eluent. The pure **13** was recovered as a colorless oil (129 mg, 62% yield).

**^1^H NMR** (300 MHz, CDCl_3_) δ(ppm) = 7.62 (d, *J* = 16.0 Hz, 1H, H5), 7.42 (d, *J* = 8.7 Hz, 2H, H2), 6.84 (d, *J* = 8.7 Hz, 2H, H3), 6.28 (d, *J* = 15.9 Hz, 1H, H6), 5.26–5.12 (m, 1H, H8), 4.36–4.07 (m, 2H, H9), 2.32 (td, *J* = 7.6, 4.5 Hz, 2H, H20), 1.72–1.49 (m, 6H, H21), 1.41–1.10 (m, 31H), 0.91–0.80 (m, 7H, H18, H28). **HRMS** (*m*/*z*) [M − H]^−^ calculated 473.3272, found 473.3270.

#### 3.2.4. Glucosylation

4-Hydroxybenzaldehyde or ethyl or dodecyl coumarate was dissolved in a 1 M solution of NaOH (1 equiv) and added dropwise to an acetone solution of 2,3,4,6-tetracetyl-α-D-glucopyranosyl bromide (1 equiv, 0.1 M). The reaction medium was stirred for 2 days at room temperature. A water addition started the precipitation process, then the precipitation was completed via solvent evaporation. The precipitate was filtered off, rinsed with water, and dried. The crude mixture was engaged without further purification in the deprotection step.

The crude product was dissolved in MeOH/Acetone (2:1), then sodium methanoate was added until reaching pH 10. After 2 h, a neutralization was performed by adding Amberlyst 15IR resin. The resin beads were filtered off and rinsed with methanol. The crude mixture was concentrated, then purified through flash chromatography over silica gel using 50:50 to 0:100 cyclohexane/EtOAc and then 90:10 EtOAc/MeOH gradient as eluent.

Ethyl glucocoumarate **17** (35% yield)

**m.p.** (°C): 157.0. **^1^H NMR** (300 MHz, Methanol-*d*_4_) δ (ppm) = 7.64 (d, *J* = 16.0 Hz, 1H, H5), 7.56 (d, *J* = 8.8 Hz, 2H, H2), 7.11 (d, *J* = 8.7 Hz, 2H, H3), 6.41 (d, *J* = 16.0 Hz, 1H, H6), 4.99–4.94 (m, 1H, H10), 4.23 (q, *J* = 7.1 Hz, 2H, H8), 3.90 (dd, *J* = 12.1, 2.1 Hz, 1H, H15), 3.79–3.62 (m, 1H, H15′), 3.51–3.34 (m, 4H, H11′14), 1.31 (t, *J* = 7.2 Hz, 4H, H9). **^13^C NMR** (Methanol-*d*_4_, 75 MHz) δ (ppm) = 169.0 (C7), 160.9 (C4), 145.7 (C5), 130.8 (C2), 129.9 (C1), 118.0 (C3), 117.1 (C6), 101.8, 78.2, 77.9, 74.8, 71.3, 62.5 (C15), 61.6 (C8), 14.6 (C9).

Dodecyl glucocoumarate **18** (30% yield)

**^1^H NMR** (300 MHz, Methanol-*d*_4_) δ (ppm) = 7.64 (d, *J* = 16.0 Hz, 1H, H5), 7.56 (d, *J* = 8.8 Hz, 2H, H2), 7.13 (d, *J* = 8.7 Hz, 2H, H3), 6.41 (d, *J* = 16.0 Hz, 1H, H6), 4.97 (dd, *J* = 5.5, 2.1 Hz, 1H, H20), 4.18 (t, *J* = 6.6 Hz, 2H, H8), 3.90 (dd, *J* = 12.1, 2.1 Hz, 1H, H25), 3.70 (dd, *J* = 12.1, 5.3 Hz, 1H, H25′), 3.54–3.33 (m, 3H, H21-24), 1.69 (p, *J* = 6.9, 6.4 Hz, 2H, H9), 1.47–1.20 (m, 19H, H10–18), 0.89 (t, *J* = 6.8 Hz, 4H, H19).^**13**^**C NMR** (Methanol-*d*_4_, 75 MHz) δ (ppm) = 169.1 (C7), 160.9 (C4), 145.7 (C5), 130.8 (C2), 129.9 (C1), 118.0 (C3), 117.1 (C6), 101.9 (C20), 78.3–71.3 (C21–C24), 65.7 (C8), 62.5 (C25), 33.1, 30.8, 30.7, 30.6, 30.5, 30.4, 29.8, 27.1, 23.7, 14.4 (C19). **HRMS** (*m*/*z*) [M + H]^+^ calculated 495.2953, found 495.2952.

#### 3.2.5. Synthesis of Dehydroxylated Alkyl Coumarates

##### Acetylation

The desired alkyl coumarate was dissolved in NaOH solution at 1 mol.L^−1^ (1 equiv), then acetic anhydride (1.5 equiv) was added dropwise at 0 °C. The reaction medium was stirred 2 h at room temperature. The precipitate was filtered, rinsed with water and dried. The acetylated alkyl coumarate was used without further purification.

##### Dihydroxylation and Deacetylation (Pathway C)

The acetylated alkyl coumarate and citric acid (0.75 equiv) were dissolved in acetonitrile/acetone/water 4:5:1 (1.0 mol.L^−1^). Then, potassium osmate (0.1 mol%) and 50% aqueous solution of *N*-methylmorpholine *N*-oxide (1.1 equiv) were added. The reaction medium turned green, was stirred until the color vanished (overnight), then quenched with 2.5 mol.L^−1^ Na_2_S_2_O_5_ solution, and extracted thrice with ethyl acetate. The organic layers were combined, washed with HCl 1 M and brine, dried over anhydrous MgSO_4_, and concentrated. Then, the crude mixture was diluted into THF (1.0 mol.L^−1^). After, piperazine (3 equiv) was added. The reaction medium was stirred overnight at room temperature, then neutralized with HCl 1 M and concentrated. The aqueous layer was extracted thrice with ethyl acetate. The organic layers were combined, washed thrice with water then brine, dried over anhydrous MgSO_4_, and concentrated. The crude mixture was subjected to flash chromatography over silica gel using 90:10 to 50:50 cyclohexane/ethyl acetate as eluent.

##### Transesterification of Ethyl Dihydroxycoumarate (Pathway D)

The general procedure for enzymatic transesterification was used starting from ethyl dihydroxycoumarate **24**.

Ethyl dihydroxycoumarate **24**, white powder (75% yield)

**m.p.** (°C): 123.4. **^1^H NMR** (300 MHz, Acetone-*d*_6_) δ (ppm) = 8.23 (s, 1H, H12), 7.25 (d, *J* = 8.1 Hz, 2H, H2), 6.78 (d, *J* = 8.2 Hz, 2H, H3), 4.86 (d, *J* = 3.7 Hz, 1H, H5), 4.37 (s, 1H), 4.11 (q, *J* = 7.1 Hz, 2H, H8), 1.17 (t, *J* = 7.1 Hz, 3H, H9). **^13^C NMR** (Acetone-*d*_6_, 75 MHz) δ (ppm) = 173.1 (C7), 157.6 (C4), 133.3 (C1), 128.8 (C2), 115.5 (C3), 76.8 (C6), 75.3 (C5), 61.3 (C8), 14.4 (C9). **HRMS** [M + NH_4_]^+^ calculated 244.1179, found 244.1177.

Yields for each compound depending of the used pathway were reported in Table 3.

Octyl dihydroxycoumarate **25**, white powder **m.p.** (°C): 61.3. **^1^H NMR** (300 MHz, CDCl_3_) δ (ppm) = 7.27 (d, *J* = 8.4 Hz, 2H, H2), 6.79 (d, *J* = 8.7 Hz, 2H, H3), 4.92 (d, *J* = 3.3 Hz, 1H, H5), 4.32 (d, *J* = 3.3 Hz, 1H, H6), 4.19 (t, *J* = 6.7 Hz, 2H, H8), 1.61 (p, *J* = 6.8 Hz, 2H, H9), 1.39-1.15 (m, 13H, H10–14), 0.88 (t, 3H, H15). **^13^C NMR** (CDCl_3_, 75 MHz) δ (ppm) = 173.0 (C7), 155.7 (C4), 132.2 (C1), 128.0 (C2), 115.5 (C3), 74.8 (C6), 74.4 (C5), 66.5 (C8), 31.9, 29.3, 28.6, 25.9, 22.8, 14.2 (C15). **HRMS** (*m*/*z*) [M − H]^−^ calculated 309.1707, found 309.1705.

Decyl dihydroxycoumarate **26**, white powder **m.p.** (°C): 75.3. **^1^H NMR** (300 MHz, CDCl_3_) δ (ppm) = 7.25 (d, *J* = 8.2 Hz, 2H, H2), 6.75 (d, *J* = 8.2 Hz, 2H, H3), 5.53 (s, 1H, H20), 4.92 (s, 1H, H5), 4.32 (d, *J* = 3.3 Hz, 1H, H6), 4.18 (t, *J* = 6.8 Hz, 2H, H8), 3.29 (s, 1H, H19), 2.85 (s, 1H, H18), 1.62 (t, *J* = 6.7 Hz, 2H, H9), 1.37–1.14 (m, 14H, H10–16), 0.88 (t, *J* = 6.6 Hz, 3H, H17). **^13^C NMR** (75 MHz, CDCl_3_) δ (ppm) = 173.0 (C7), 155.7 (C4), 132.0 (C1), 127.9 (C2), 115.5 (C3), 74.9 (C6), 74.4 (C5), 66.5 (C8), 32.0, 29.7, 29.6, 29.4, 29.3, 28.6, 25.9, 22.8, 14.25 (C17). **HRMS** (*m*/*z*) [M − H]^−^ calculated 337.2020, found 337.2023.

Dodecyl dihydroxycoumarate **27**, white powder **m.p.** (°C): 85.5. **^1^H NMR** (300 MHz, CDCl_3_) δ (ppm) = 7.25 (d, *J* = 8.3 Hz, 2H, H2), 6.75 (d, *J* = 8.5 Hz, 2H, H3), 5.46 (s, 1H, H22), 4.93–4.87 (m, 1H, H5), 4.32 (dd, *J* = 5.9, 3.3 Hz, 1H, H6), 4.18 (t, *J* = 6.7 Hz, 2H, H8), 3.27 (d, *J* = 6.1 Hz, 1H, H21), 2.82 (d, *J* = 5.9 Hz, 1H, H20), 1.62 (q, *J* = 6.8 Hz, 2H, H9), 1.35–1.19 (m, 18H, H10–18), 0.88 (t, *J* = 6.6 Hz, 3H, H19).^**13**^**C NMR** (CDCl_3_, 75 MHz) δ (ppm) = 173.0 (C7), 155.7 (C4), 132.0 (C1), 127.9 (C2), 115.5 (C3), 74.9 (C6), 74.4 (C5), 66.6 (C8), 32.1, 29.8, 29.7, 29.6, 29.5, 29.3, 28.6, 25.9, 22.8, 14.3 (C19). **HRMS** (*m*/*z*) [M − H]^−^ calculated 365.2333, found 365.2333.

Tetradecyl dihydroxycoumarate **28**, white powder **m.p.** (°C): 90.4.^**1**^**H NMR** (300 MHz, CDCl_3_) δ (ppm) = 7.27 (d, *J* = 8.2 Hz, 2H, H2), 6.78 (d, *J* = 8.5 Hz, 2H, H3), 4.92 (d, *J* = 3.2 Hz, 1H, H5), 4.32 (d, *J* = 3.2 Hz, 1H, H6), 4.19 (t, *J* = 6.7 Hz, 2H, H8), 1.61 (q, *J* = 6.9 Hz, 2H, H9), 1.33–1.15 (m, 23H), 0.87 (t, 3H, H21).^**13**^**C NMR** (CDCl_3_, 75 MHz) δ (ppm) = 173.0 (C7), 155.6 (C4), 132.2 (C1), 128.0 (C2), 115.5 (C3), 74.8 (C6), 74.4 (C5), 66.6 (C8), 32.1, 29.8, 29.8, 29.7, 29.6, 29.5, 29.3, 28.6, 25.9, 22.8, 14.3 (C21). **HRMS** (*m*/*z*) [M − H]^−^ calculated 393.2646, found 393.2648.

2-butyl octyl dihydroxycoumarate **29**, white powder **m.p.** (°C): 61.1.^**1**^**H NMR** (300 MHz, CDCl_3_) δ (ppm) = 7.23 (d, *J* = 8.2 Hz, 2H, H2), 6.74 (d, *J* = 8.2 Hz, 2H, H3), 5.66 (s, 1H, H22), 4.90 (dd, *J* = 6.1, 3.2 Hz, 1H, H5), 4.33 (dd, *J* = 6.0, 3.3 Hz, 1H, H6), 4.10 (d, *J* = 5.8 Hz, 2H, H8), 3.34 (d, *J* = 6.2 Hz, 1H, H20), 2.87 (d, *J* = 6.3 Hz, 1H, H21), 1.68–1.51 (m, 1H, H9), 1.26 (s, 16H, H10–14, H16–18), 0.97–0.76 (m, 6H, H15, H19).^**13**^**C NMR** (75 MHz, CDCl_3_) δ (ppm) = 173.1 (C7), 155.8 (C4), 131.9 (C1), 127.9 (C2), 115.5 (C3), 74.9 (C6), 74.4 (C5), 69.2 (C8), 37.3 (C9), 31.9, 31.2, 30.8, 29.7, 29.0, 28.9, 26.8, 26.7, 23.1, 22.8, 14.2, 14.2 (C15, C19). **HRMS** (*m*/*z*) [M − H]^−^ calculated 365.2333 found 365.2334.

### 3.3. Biological Assays

#### 3.3.1. Growth Conditions of Plant and Fungus Material

*Arabidopsis thaliana* ecotype Col-0 (Arabidopsis) was used in all experiments. The seedlings were grown in Gramoflor brand potting soil (range 20/80) and placed into a growth chamber at 21 °C (photoperiod 12/12) for 2 weeks. Then, the seedlings were isolated in pots, with each pot containing one seedling. The plants were used at the age of 4-to-6 weeks for all experiments.

*Brassica napus* (rapeseed) was used in all experiments. The seedlings were grown in Gramoflor brand potting soil (range 20/80) and placed into a growth chamber at 19 °C (photoperiod 14/10). The cotyledons were used when fully developed at the age of 2 weeks.

*Botrytis cinerea* strain 630 and *Sclerotinia sclerotiorum* 51 fungi were grown on Potato Dextrose Agar Petri dishes in a growth chamber at 19 °C in dark for 2 weeks.

#### 3.3.2. In Vitro Antifungal Assay

The molecules of interest were mixed with Potato Dextrose Agar (3.9 g/L) to obtain a 100 µM concentration and placed into Petri dishes. Conidia of *B. cinerea* were collected from 10-day-old culture plates with 4 mL of growth culture Potato Dextrose Broth filtered to remove mycelia and counted. A drop of 5 µL at 10^5^ conidia/mL was deposited on the center of each Petri dish. A plug of 5 mm diameter of *S. sclerotiorum* was collected from 10-day-old culture plates and was deposited on the center of each Petri dish. Those Petri dishes were placed in a growth chamber at 19 °C in dark for 3 days. Then, pictures were taken using colony counter Scan500 (Interscience). The pictures were studied using software ImageJ 1.53e to determine the extent of the mycelium. Relative inhibition was calculated with the following formula [44]:$$RI\;\left(\%\right)=\left(\frac{average\;surface\;of\;the\;control-average\;surface\;of\;the\;sample}{average\;surface\;of\;the\;control}\right)\times 100$$

#### 3.3.3. *In Planta* Ion Leakage Assay

Ion leakage assays were performed on 4-to-6-week-old Arabidopsis plants or 2-week-old cotyledon rapeseed plants cultured in soil, as previously described. Two leaf discs of 8 mm diameter were incubated in ultra-purified water for 2 h in each well of a 24-well plate (Falcon). Water was removed from every well and replaced with fresh ultra-purified water and the corresponding molecule at 100 µM, or only water for control. Ion leakage is determined by conductivity measurements (three replicates for each treatment were then conducted using a B-771 LaquaTwin (Horiba) conductivity meter at 0 h post-inoculation (hpi), 24 hpi, and 48 hpi.

#### 3.3.4. Plant Protection Assays

The conidia of *B. cinerea* were collected from 10-day-old culture plates with 4 mL of growth culture Potato Dextrose Broth, filtered to remove mycelia, and counted. 4-to-6-week-old *Arabidopsis* plants received 3 to 4 sprays of the molecules of interest at 100 µM final concentration, or water for control, and were placed in growth chambers at 20 °C, 12 h of light, 12 h of darkness, and 55% of humidity. Three days later, 3 to 4 leaves of these *Arabidopsis* plants were excised and placed on Petri dishes containing Agar 7%. A drop of 5 µL at 10^5^ conidia/mL was deposited on the center of each leaf. Petri dishes were placed in growth chamber at 19 °C for 3 days. Pictures of the Petri dishes were taken with the Scan500 device. Necrotic areas were measured using ImageJ software 1.53e. 

## 4. Conclusions

Sustainable Knoevenagel-Doebner condensation or enzymatic transesterification were employed to establish the library of *p*-coumaric esters with high yield. Out of the 17 *p*-coumaric acid derivatives synthesized, the dihydroxylated ones were found the most potent for both the direct inhibition growth of the fungi and plasma membrane destabilization. They were selected for plant protection assays and have also been proven efficient by reducing the necrotic areas of infected leaves. To summary, decyl dihydroxycoumarate was found to be the most active compound, proving that a balance between chain length and hydrophilicity is required to achieve efficient antifungal activity and insertion into the plant plasma membrane. Further experiments are required to confirm that these molecules family could be potential candidates to be used in biocontrol strategies for crop protection against fungal diseases.

## Data Availability

Not applicable.

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
