# Peer review of "Chemo-Enzymatic Synthesis and Biological Assessment of p-Coumarate Fatty Esters: New Antifungal Agents for Potential Plant Protection"

_molecules, 2023, doi:10.3390/molecules28155803_

Round 1
Reviewer 1 Report
It is always to read a manuscript like this. The authors synthesized a library of p- 14 coumaric-based compounds and screened their functions in controlling phytopathogens. I do not have too many problems accepting this manuscript, but I still think the discussion section of this article is not comprehensive. Honestly, I agree that a combination of the data and discussion can make this manuscript more readable, however, I strongly suggest an extension of the discussion part to give more information on the future usage or application or even economic cost of these molecules.
Author Response
We thank the reviewer for acknowledging the value of our work. As suggested, we have enhanced the results and discussion part by adding a “discussion and perspectives” section to go further into the potential for these compounds.
Reviewer 2 Report
Dear Authors,
congratulations on the manuscript. The topic of the manuscript is important, but the following additions and amendments are recommended:
The introduction is a bit short compared to the scope of the draft article. Please supplement this with additional references. The format of the References chapter is appropriate.
Where possible, it would be better to use a tabular form instead of textual parts in Material and method. It would be more transparent if you could solve this.
Is the Appendix part of the manuscript? Not a separate file?
May you have a nice day at work!
The English language is adequate, only minor corrections are needed.
Author Response
We thank the reviewer for acknowledging the value of our work.
As requested, the introduction has been revised to introduce some additional information.
A table has been added in the material section to clarify the supplier of every single product used in this work. In the method section, descriptions of the molecule have not been changed as it is the formal way to present it. However, tables were added to facilitate the comparison of yields between the various pathways.
Appendix is placed at the end of the manuscript in the template provided by the editorial office.
Reviewer 3 Report
In this paper, the contents include: synthesis of a library of p-coumaric-based compounds, antifungal activity evaluation for two pathogens, ion leakage test, and protective effect on Arabidopsis. The synthesized coumarate fatty esters are of interest. 100 µM of these compounds having good antifungal activity against Botrytis cinerea and Sclerotinia sclerotiorum. Decyl dihydroxycoumarate was found to be the most active compound, which has protective effect on Arabidopsis.
Revision comments:
1. The title is not fit. No experimental data to support the crops elicitation. The title must be corrected.
2. the positive control is needed in the biological assessments.
3. EC50 for decyl dihydroxycoumarate is required.
Author Response
1. Title has been updated accordingly
2. Controls have been added in the figures
3. EC50 of decyl dihydroxycoumarate has been determined at 25 µM and added in the discussion.
Round 2
Reviewer 3 Report
This paper has been improved well according to the referees` comment. It is recommended for publication.
a little problem
Scheme 2, please replace the carbon number in the appropriate place.